# BIASED CAUSAL STRENGTH JUDGMENTS IN HUMANS AND LARGE LANGUAGE MODELS

**Anita Keshmirian**[1,2,3]**, Moritz Willig**[4]**, Babak Hemmatian**[5,6]**, Ulrike Hahn**[2,7]**,
Kristian Kersting**[4,8,9,10] **& Tobias Gerstenberg**[11]

[1] Fraunhofer Institute for Cognitive Systems (IKS) - Munich, Germany
[2] Munich Center for Mathematical Philosophy (MCMP) - LMU, Munich, Germany
[3] Forward College, Berlin, Germany
[4] Computer Science Department, Technical University of Darmstadt, Germany
[5] Beckman Institute for Advanced Science and Technology, University of Illinois
    Urbana-Champaign, USA
[6] Center for Brain, Biology and Behavior, University of Nebraska Lincoln, USA
[7] Department of Psychology, Birkbeck University, London, UK
[8] Centre for Cognitive Science, Technical University of Darmstadt, Germany
[9] Hessian Center for Artificial Intelligence (hessian.AI), Germany
[10] German Research Center for Artificial Intelligence (DFKI), Germany
[11] Department of Psychology, Stanford University, USA

## ABSTRACT

Causal reasoning is a critical aspect of human cognition and artificial intelligence (AI), which plays a prominent role in understanding the relationships between events. Causal Bayesian Networks (CBNs) have been instrumental in modeling such relationships, using directed, acyclic links between nodes in a network to depict probabilistic associations between variables. Deviations from these graphical models' edicts would result in biased judgments. This study explores one such bias in the causal judgments of humans and Large Language Models (LLMs) by examining two structures in CBNs: Canonical Chain (A→B→C) and Common Cause (A←B→C) networks. In these structures, once the intermediate variable (B) is known, the probability of the outcome (C) is normatively independent of the initial cause (A). However, studies have shown that humans often ignore this independence. We tested the mutually exclusive predictions of three theories that could account for this bias ($N = 320$). Using hierarchical mixed-effect models, we found that humans tend to perceive causes in Chain structures as significantly stronger, providing support for only one of the hypotheses. This increase in perceived causal power might reflect a view of intermediate causes as more reflective of reliable mechanisms, which could, in turn, stem from our interactions with the world or the way we communicate causality to others. LLMs are primarily trained on language data. Therefore, examining whether they exhibit similar biases in causal reasoning can help us understand the origins of canonical Chain structures' perceived causal power while also shedding light on whether LLMs can abstract causal principles. To investigate this, we subjected three LLMs, `GPT3.5-Turbo`, `GPT4`, and `Luminous Supreme Control`, to the same queries as our human subjects, adjusting a key 'temperature' hyperparameter. Our findings show that, particularly with higher temperatures (i.e., greater randomness), LLMs exhibit a similar boost in the perceived causal power of Chains, suggesting the bias is at least partly reflected in language use. Similar results across items suggest a degree of causal principle abstraction in the studied models. Implications for causal representation in humans and LLMs are discussed.

## 1 INTRODUCTION

Representations of causal structure guide our reasoning and shape our interpretations of reality. For instance, keeping all else constant, people provide different judgments of a causal Chain, a sequence

of causally related events that result in an outcome, than a Common Cause structure where an underlying factor gives rise to multiple effects (Rehder, 2014). Would such a difference in structure - all else being equal - lead to systematic differences in the perceived likelihood of a cause (i.e., its causal strength)?

## 1.1 CAUSAL BAYESIAN NETWORKS (CBNS)

Causal Bayesian Networks (CBNs) provide a common approach to causal structure representation, which has been fruitfully applied to similar questions (Pearl, 2009). CBNs are graphs that depict probabilistic inter-dependencies between variables. The variables (called "nodes") are interconnected through directed arrows (called "edges") into a-cyclic structures, indicating their probabilistic associations.

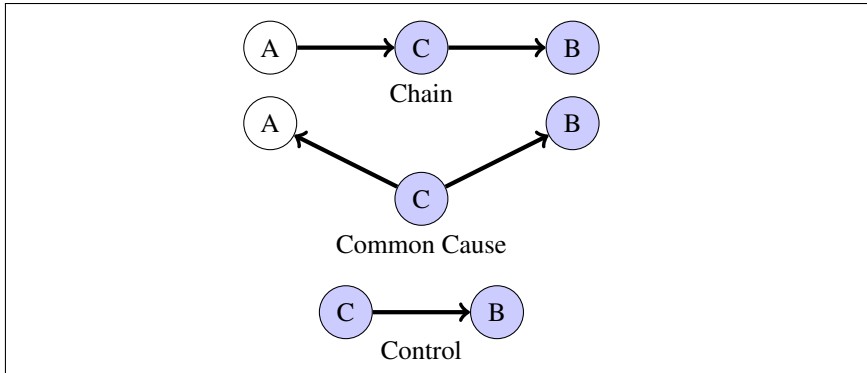

Figure 1: The joint probability $P(A, B, C)$ in canonical Common Cause and Chain Causal Bayesian Networks is identical. Given $C$, the causal strength of $C \to B$ is independent of the network structure.

Bayes Theorem provides a prescriptive framework for updating beliefs based on evidence according to such networks in a rational and consistent way, a deviation from any of its axioms leading to demonstrably sub-optimal reasoning. Two well-studied CBNs that we will focus on are shown in figure 1: three-node Chains and Common Cause networks. The joint probability, the probability that the events represented by all three nodes happen in an instance of the causal structure, is equal for the two graphs (see equation 1). This "equivalence class" means that a given dataset would have the same likelihood under both structures, indicating that they cannot be differentiated solely based on observational data. Therefore, any systematic differences in our intuitions of causal strength across them are not due to the networks' overall likelihood.

$$P(A, B, C) = P(B \mid C)P(C \mid A)P(A) = P(A \mid C)P(B \mid C)P(C) \tag{1}$$

But more central to our research question is the notion of conditional independence that leads to this equivalence: the probability of B should not depend on A if we know C. In other words, for a given value of C, the likelihood of C→B, and therefore the causal strength of C for bringing B about, should be the same for the Chain and Common Cause networks in figure 1.

## 1.2 HUMANS AND THE INDEPENDENCE ASSUMPTIONS

Humans systematically violate the independence assumptions in their causal judgments (Mayrhofer et al., 2008; Park & Sloman, 2013; Rehder & Burnett, 2005; Rehder, 2014). Recently, the direct scope of a cause, i.e., the number of distinct effects generated directly by it, has been studied as a source of perceived causal strength (or lack thereof) not predicted by Bayesian theory (Sussman & Oppenheimer, 2020; Zemla et al., 2017). In Chain A→C→B, the direct scope of C is one, less than the node's scope in the Common Cause structure A←C→B, which is two. Sussman and Oppenheimer (Sussman & Oppenheimer, 2020) argued that, depending on the valence of a target

effect (B in this case), a broader scope might enhance or diminish perceived causal strength. For positive effects ("boons"), broader scope increases perceived causal strength: a drug preventing three negative symptoms is stronger than one that prevents a single symptom. When effects are negative ("banes"), however, a broader scope leads to lower perceived strength. Like most prior research, Sussman and Oppenheimer (Sussman & Oppenheimer, 2020) limited their comparison to a single structure type: Common Cause networks, with two-variable direct causation as the baseline.

Stephan et al. (2023) found that when scenarios were abstract enough to eliminate prior domain beliefs in participants (e.g., an alien on Mars eating a red crystal that induces/prevents three vs. one attribute(s)), the effect of scope was uniform across positive and negative outcomes. They instead found a "dilution effect": in causes with broader scopes (three effects in their experiments rather than one), a singular "source" of causal strength is seen as distributed and, therefore, "diluted" across the multiple effects regardless of effect valence.

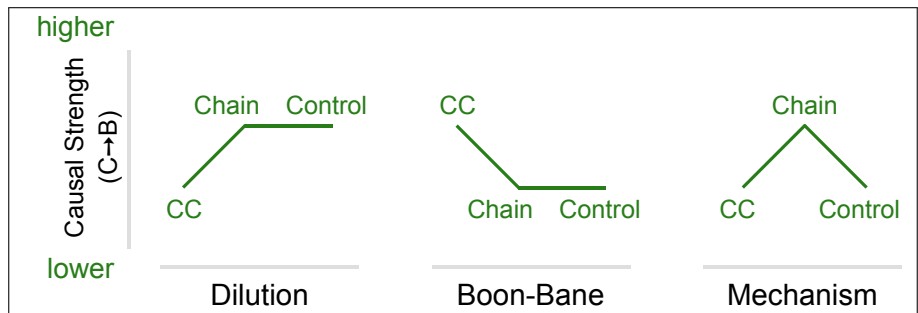

Figure 2: Pattern predictions for the effect of causal structure on perceived causal strength estimates across conditions. The predictions of the different theories for networks with negative-valence contents are illustrated.

Park & Sloman (2013) uncovered another systematicity in people's responses that may be relevant to perceived causal strength in these cases: Subjects' deviations from experimenter-provided causal structures and their subsequent apparent violations of normative reasoning always revolved around mechanisms and could be changed with mechanism-directed instructions. For instance, based on whether the same mechanism accounted for the two effects in a Common Cause network or the mechanism for each effect was distinct, perceptions of the causes' influence differed significantly. This finding adds to a large body of literature in the cognitive sciences demonstrating the centrality of mechanisms to causal reasoning in ways that do not always track normative predictions (Johnson & Ahn, 2017). For instance, Zemla and colleagues (Zemla et al., 2017) found that mechanistic information subverts a preference for simplicity in explanations, with subjects going so far as identifying enough causes to make the effect seem inevitable. From a more normative standpoint, Russo and Williamson (2007) have emphasized the importance of mechanistic evidence in discerning alternative non-causal explanations, such as confounding, bias, or chance, which may lead to misleading associations. If a plausible mechanism to explain the correlation is absent, the association will likely be merely coincidental (Russo & Williamson, 2007). If we adopt the view that mechanistic relevance is seen as a link in the causal chain connecting the input (the initial cause) and the output (the final effect) (Menzies, 2012), C may be perceived in the Chain network of figure 1 (i.e., A→C→B) as the mechanism that explains A's impact on B. If so, the perceived causal strength of C on B in A→C→B would be higher than in a control condition C→B, as the chain would be seen as more likely to represent a mechanistic rather than a merely correlational relation. In the Common Cause network, on the other hand, the mechanism is vague without elaboration ( figure 2: Mechanism). Therefore, if people accept the provided networks as the ground truth (Rehder, 2014) but base their causal strength judgments on other structural features like the provision of a mediating process, we would expect the Chain condition to have higher causal likelihood ratings. Figure 2 shows how three experimental conditions would allow this hypothesis to be evaluated alongside the accounts of Boon-Bane theory (Sussman & Oppenheimer, 2020) as well as the Dilution theory (Stephan et al., 2023).

### 1.3 LARGE LANGUAGE MODELS AND THE INDEPENDENCE ASSUMPTIONS

A key question in cognitive science concerns the role of language in causal reasoning, with important implications for the causal capabilities of large language models (LLMs) (Binz & Schulz, 2023; Willig et al., 2022). Human beings can communicate causal information via language, but they also develop an understanding of causality through interactions with the world. Carrying out exact computational operations internally, LLMs can — in theory — perform perfect normative reasoning. However, trained almost exclusively on human textual data, we expect LLMs to pick up on biases that are reflected in language use but not those only learned through experience. To see whether LLMs also perceive intermediate causes in canonical Chain structures as stronger, we collect and compare LLM answers with the distribution of human judgments (cf. figure 5).

If LLMs, trained predominantly on linguistic data, acquire similar biases in their causal reasoning, it would stand to reason that such elements of human judgments are at least partially imparted through language. Such a finding would also contribute to an ongoing debate within the AI community about whether LLMs grasp causality or merely echo causal language without comprehension Zečević et al. (2023). Many researchers have recently taken the stance that current LLMs are not able to generalize causal ideas beyond their training distribution and/or without strong user-induced guidance (Kıcıman et al., 2023; Jin et al., 2023). But if a preference for canonical Chain over Common Cause structures emerges across items in experiments with LLMs, that would provide some evidence that LLMs suffer from the influence of human bias for causal principle abstraction.

## 2 HUMAN EXPERIMENT

### 2.1 METHOD

#### 2.1.1 DESIGN

We manipulated causal structures between subjects to minimize task demands. We limited our materials to structures with three nodes to minimize working memory demands also because none of the general explanations we evaluated demanded degrees higher than two. Stephan et al. (2023) highlighted intuitive familiarity with the variables as a potential explanation for finding different results than Sussman & Oppenheimer (2020). To account for this possibility, we included an adapted version of Stephan et al. (2023) Alien scenario, meant to preclude prior knowledge of the causal relations. Our two other items represented more intuitive domains: a widely used causal setup about the Economy (adapted from (Rehder, 2014)), and a novel scenario about Sex Work Criminalization that represented more prescriptive causal reasoning. The C→B relation was always presented as probabilistic (C can lead to B) to prevent ceiling effects seen in a pilot study. Because Sussman and Oppenheimer's (2020) account deviates from the Dilution predictions only when target effects are negatively valenced, node B in figure 1 was negatively valenced in all scenarios to provide better experimental contrast. Following (Rehder, 2014), we instructed participants to consider all presented relations as "single sense," meaning that only the presence of the event represented by a node has an impact on its effect(s).

#### 2.1.2 NETWORK STRUCTURE

For the main manipulation, each participant was assigned to one of the following conditions at random: 1. Chain, where A generates C, which in turn generates B (A→C→B), 2. CC, a Common Cause network where C separately generates both A and B (A←C→B), 3. Control, a baseline for (1) and (2) in which A is not included and C generates B (C→B). For instance, participants in the Chain condition were presented with scenarios such as: "Criminalizing sex work leads to greater profits for criminal organizations, which can then lead to higher gender-based crime rates." In contrast, those in the Control condition encountered scenarios like: "Greater profits for criminal organizations can lead to higher gender-based crime rates."

To examine whether generative causation behaves differently than preventative causal power in these scenarios (Walsh & Sloman, 2011), we compared two additional conditions: 4. CC(P): a Common Cause network where C prevents A and separately inhibits B (A←C→B), 5. Control(P): a baseline for (3) in which A is not included and C inhibits B (C→B). A random pairing of scenarios was created for within-subject manipulations and then counterbalanced across participants. Scenarios

were individually presented on the screen in randomized order. Demographics followed the last vignette.

### 2.1.3 DEPENDENT MEASURE

The dependent measure was always the likelihood of the effect B if C had been present. For example, participants in the aforementioned condition were presented with the statement: "Criminalizing sex work leads to greater profits for criminal organizations, which can then lead to higher gender-based crime rates." Subsequently, participants were asked to evaluate the likelihood of the statement: "How likely is it that greater profits for criminal organizations lead to higher gender-based crime rates?" The C→B causal relationship was scrutinized as it remained constant across the network types under examination.

### 2.1.4 PARTICIPANTS

We collected pilot data from 100 participants to calculate the needed sample size using the simulation-based method (DeBruine & Barr, 2021). A target sample size was predetermined using a Monte Carlo simulation via the `SIMR` package in *R* (Green & MacLeod, 2016). We estimated the input parameters of our simulation to determine the sample size to have 90 % power to detect the main effect of causal structure. Our final estimated sample size was 300.

Three hundred and twenty-nine participants were recruited through Prolific Academic and compensated at the average rate of $10/hr. The ($\sim 10\%$) increase in sample size over the simulation was meant to offset anticipated losses in the degrees of freedom due to inattentive subjects.

We used a data-driven Mahalanobis Distance measure (Leys et al., 2018) to identify non-human participants and inconsistent or inattentive responses. This step resulted in excluding 6 participants. We replicated the main results, including those who failed the Mahalanobis exclusion criterion. Since our secondary attention check excluded a third of the participants, to preserve power, we limited our exclusion to the data-driven approach explained above. However, the key results were replicated after excluding subjects who failed the secondary check (see OSF for details). The final sample of US and UK residents (122 males, 195 females, 5 choosing the "non-binary" option) had an average age of 37.29 years ($SD = 13.02$, range: 18 to 76).

### 2.2 REVIEW OF HYPOTHESES

Figure 2 summarizes the target contrasts examined across theories and CBN types. All the predictions listed except for the Mechanism explanation are included in the OSF preregistration.

Dilution theory (Stephan et al., 2023) predicts that causal strength would be reduced if a cause has two direct effects rather than one, regardless of other structural features and the effect's valence. Therefore, less causal strength is expected in CC and CC(P) conditions where C has two effects, compared with Chain and Control conditions where it contributes to only one effect. Given that target effects all have negative valence, the Bane-Boon Theory Sussman & Oppenheimer (2020) predicts the opposite pattern: CC and CC(P) would be expected to receive higher ratings due to their wider direct scopes than the Chain and Control conditions. The Mechanism hypothesis, on the other hand, expects higher causal strength in the Chain condition where subjects may regard C as a mechanistic cause, which is preferred over covariational ones (Ahn & Bailenson, 1996). Since no intermediate causes exist in Control and Common Cause conditions, this notion predicts no difference between them.

### 2.3 RESULT

Generalized mixed-effects models appropriate for our design's hierarchical structure were used. Since the dependent variable was on a 100-point scale, we employed linear mixed-effect models through the `LME4` package in *R* (Bates et al., 2015). Structure (Chain vs. Common Cause vs. Control) was a fixed effect, while participants and scenarios served as random effects along with the 'maximal' slopes advocated by prior research (Barr et al., 2013). figure 3 shows the experimental results. Structure's main effect was significant ($b = 8.78, \text{SE} = 3.29, z = 2.67, p = .01$, two-tailed test). However, the direction was in contrast to the predictions of Boon-Bane and

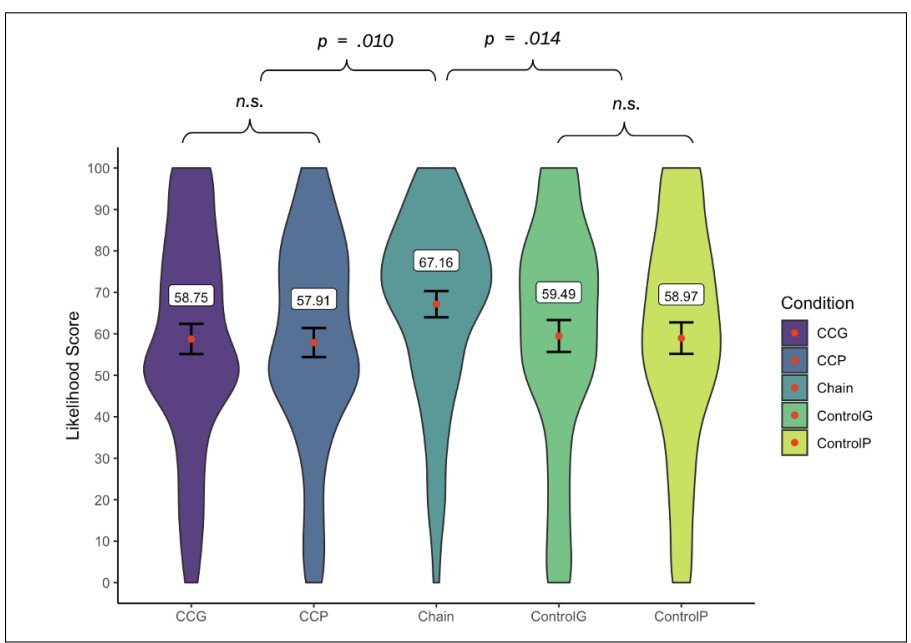

Figure 3: Distribution of responses over conditions. The likelihood in the Chain Condition (middle row) is significantly higher (i.e., higher likelihood score) than in the rest of the conditions. However, there is no difference between Control (CP; CG) and Common Cause (CCP; CCG) networks. Error bars show 95% confidence intervals.

Dilution accounts, aligning instead with a view of $C$ as a mechanistic cause. To better interpret the results, we compared Chains to Control and Common Cause networks in separate models. Random parameters were included as before. We calculated contrasts over estimated marginal means using `emmeans` (Lenth et al., 2018). Pairwise contrasts with the Tukey adjustment for multiple comparisons showed higher ratings for causes in Chains than Control ($b = 7.48, \text{SE} = 3.01, z = 2.48, p = .014$, two-tailed test: $BF_{10} = 0.86$) and Common Cause ($b = 8.74, \text{SE} = 3.34, z = 2.61, p = .010$; two-tailed test; $BF_{10} = 0.71$). No significant difference was observed between Control and Common Cause conditions in Generative ($b = 2.26, \text{SE} = 3.33, z = .67, p = .91$, two-tailed test) or Preventive ($b = 1.48, \text{SE} = 3.28, z = .47, p = .96$, two-tailed test) conditions (figure 3). A similar pattern across items suggests that prior familiarity with the domain had little influence (figure 4). Since the lack of Dilution is based on a null effect, a Bayesian mixed-effect analysis was performed to determine its reliability using the `brms` package in *R* (Bürkner, 2017). The Bayesian Mixed Effect model confirmed the null effect in both Generative ($BF_{10} = 0.96, CI_{95} = [-1.5, 2.21]$) and Preventative conditions ($BF_{10} = 1.5, CI_{95} = [-1.93, 1.74]$). To further ascertain that the effect size we observed was close to zero, an equivalence test was performed using the `TOSTER` package (Lakens, 2017). The equivalence test confirmed that the distributions of likelihood scores in Common Cause vs. Control are equivalent ($z = 8.9, p < 0.001$, two-tailed test), further confirming the null result.

## 2.4 DISCUSSION

We compared three explanations for changes in perceptions of causal strength based on network structure not predicted by normative theory. We limited our comparison to 3-node Chain and Common Cause structures as canonical network formats for which the hypotheses offered mutually exclusive predictions. We found no evidence of causal strength dilution for nodes with more direct effects (Stephan et al., 2023) or an increase in perceived strength given negatively valenced material (Sussman & Oppenheimer, 2020). However, Causal Chains of the canonical form studied received significantly higher likelihood ratings than Common Cause networks across the board. Of the three possibilities examined, this finding is consistent with the Mechanism hypothesis (figures 2-3). This

hypothesis suggests that an intermediate cause in a canonical chain may be seen as a reliable mechanism, which is preferred over a covariational cause (Ahn et al., 1995; Park & Sloman, 2013). In our networks, the influence of C over the final effect is not only more direct than the initial cause but, if seen as a mechanistic explanation, may be deemed more generalizable to other relevant instances (Johnson & Ahn, 2017). Intermediate causes were missing in the Common Cause networks, precluding such conjectures.

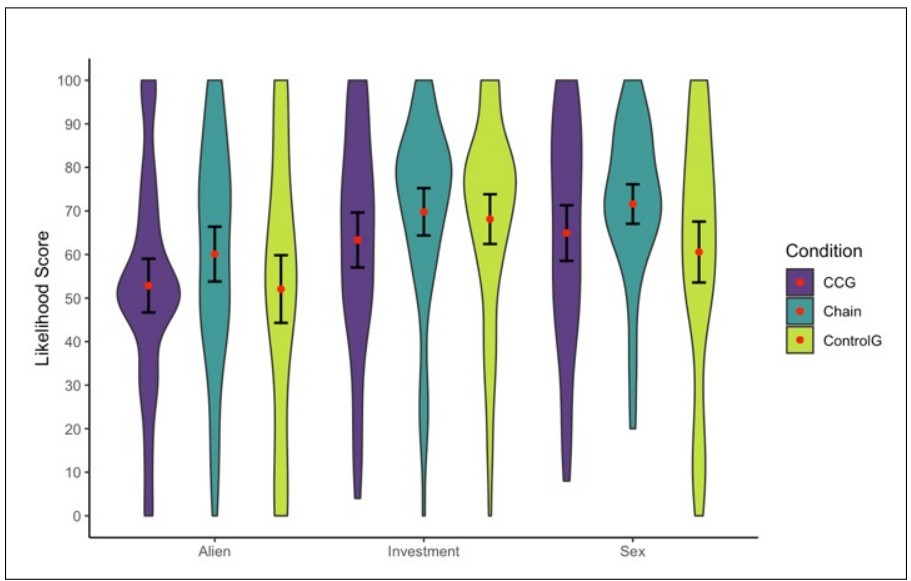

Figure 4: Distribution of responses across conditions for the various items. The likelihood in Chains (in Green) is higher than the rest of the generative conditions across all items. Error bars show 95% confidence intervals.

## 2.5 LARGE LANGUAGE MODELS

To examine whether the bias observed in human judgments is reflected in LLM responses, we compared our behavioral data with the responses generated by three recent generative models: GPT3.5-Turbo (OpenAI, 2022), GPT4 (OpenAI, 2023), and Luminous Supreme Control (Aleph Alpha, 2022). Instead of using default hyperparameter values, we systematically adjusted the temperature settings of these models to explore whether deterministic (zero temperature) or increasingly non-deterministic behavior more closely matches human reasoning in this task.

### 2.5.1 METHODS

We queried the models for all conditions and items with temperatures ranging from $0.0$ (deterministic) to $2.0$ (highly non-deterministic; maximum value permitted by the APIs) in $0.1$ steps. The temperature for the *Luminous* model was scaled by $0.5$ to match the GPT models. Surprisingly, *Luminous* did not show fully deterministic behavior at zero temperature (cf. figure 5; Wasserstein distance is non-zero). For every condition-temperature-model combination, we sampled $100$ answers for the same query. The queries were highly similar to the scenarios used with humans, except for the use of a template that encouraged the models to give a number between 0 and 100 as a response. The prompt template was structured as follows: *This is a hypothetical question.* `<insert human scenario here>`*. Given the information above, in your opinion, on a scale of 1 to 100, how likely is it that* `<insert the causal statement examined>`*? Please respond only with one single number and no text.*

## 2.6 STATISTICAL ANALYSIS

We used hierarchical mixed-effect models to discern patterns in the responses. To quantify the similarity between human and AI judgments, we computed Earthmover's distance (EMD, also known as Wasserstein distance (Vaserstein, 1969; Kantorovich, 1960), defined between two distributions u and v as follows:

$$\text{EMD} = \inf_{\pi \in \Gamma(u,v)} \int_{\mathbb{R} \times \mathbb{R}} |x - y| d\pi(x, y) \tag{1}$$

Where $\Gamma(u, v)$ is the set of all probability distributions with marginals u and v. A key property of EMD on categorical distributions is its invariance to overall shifts of the distribution, which lets us focus on the similarity of answer frequencies, between humans and models. More details about the measure's calculation, its implementation through Python's `SciPy` package, code, LLM responses, and evaluations are available at https://github.com/MoritzWillig/biasedCausalStrengthJudgments/.

Other commonly used f-divergent metrics (such as Kullback–Leibler (Kullback & Leibler, 1951) divergence) do not apply to our comparison, as they vary with the overall mean of the distribution and break down in our scenario of sparsely populated distributions. For instance, with zero temperature, a single entry receives all the probability mass, rendering KL divergence meaningless. Nonetheless, to ensure comprehensiveness, we evaluated entropy and KL divergence and have included the results in the repository. EMD estimates help us gauge the extent to which LLMs' reasoning aligns with human judgment and whether their judgments are as diverse or predictable as those of humans. Note, however, that there are qualitative differences between sampling from N different human participants and sampling N answers from the latent distribution of a single LLM with varying temperatures. Despite this distinction, we find that, given certain temperature settings, human and LLM distributions tend to approach each other.

The mixed effects model with the fixed factor of Condition (Chain, Common Cause, Control) and the random slope of Model (GPT4, GPT3.5, Luminous) highlighted a bias in the LLMs similar to the one we found in humans, attributing greater causal strength to the intermediate cause in canonical causal Chains ($\mu = 67.59, \text{SD} = 20.2$) than to the corresponding node in a Common Cause structure ($\mu = 64.89, \text{SD} = 19.8; b = 3.02, \text{SE} = .27, z = 10.89, p < .0001$, two-tailed test) and the Control condition ($\mu = 55.05, \text{SD} = 17.89; b = 12.15, \text{SE} = .28, z = 42.6, p < .0001$, two-tailed test). Behavior was particularly similar to humans when the temperature parameter surpassed 1.

Figure 5 shows the effect of switching from a Common Cause to a Chain network on dependent scores for each model using Wasserstein Distance averaged per scenario. For comparison, we indicate the distance observed in the human data, representing generally higher ratings for Chain than Common Cause structures, with a red line. For GPT-3.5-Turbo and Luminous, low temperatures correspond to little preference for either condition, but the distance between the condition distributions increases afterward. Both models eventually converge towards a randomly sampled uniform distribution with distance values below the human reference. The most recent LLM, GPT-4, starts with zero distance, but the preference for Chains increases with higher temperature. Generally speaking, temperatures $> 1.0$ match the human data best on average, while too high of a temperature value $> 1.9$ induces too much variance. With temperatures between 1.0 and 1.9, the observed preference for Chains is remarkably similar to that observed in humans across all three models.

## 2.7 RESULT

## 2.8 GENERAL DISCUSSION

We examined the effect of network structure on causal strength judgments in humans and Large Language Models (LLMs). Human participants and multiple LLMs - GPT3.5-Turbo (OpenAI, 2022), GPT4 (OpenAI, 2023), and Luminous Supreme Control (Aleph Alpha, 2022) - considered intermediate causes in chains to be more potent than causes in simple C→B networks, or those with multiple independent direct effects (i.e., Common Cause), representing a violation of normative Bayesian reasoning. Varying LLM hyperparameters, we found the closest match for the human bias

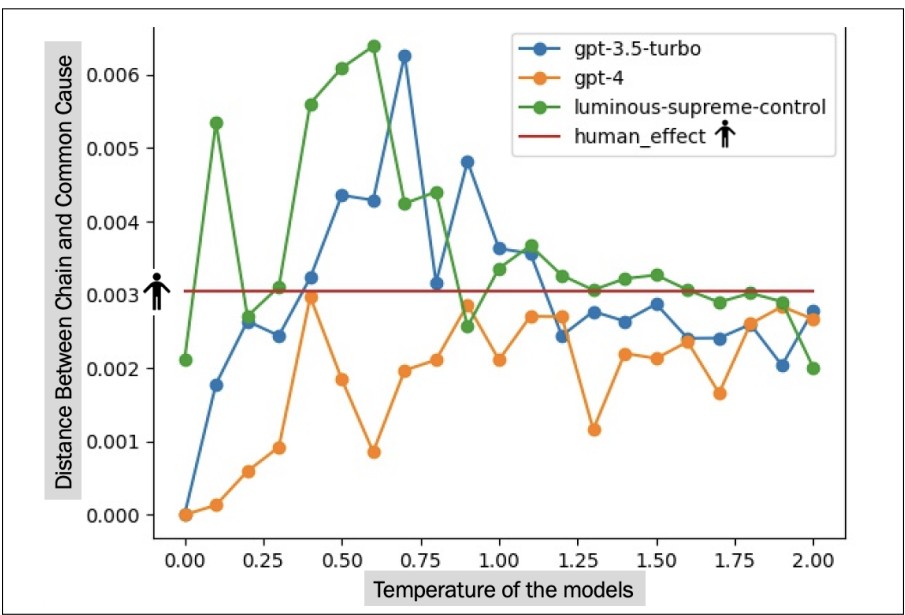

Figure 5: Average effect of switching from CC to CHAIN condition. The effect is measured as the Wasserstein distance between the Chain and Common Cause distributions for various 'temperature' values.

in variants with higher temperatures, i.e., those incorporating more randomness into the AI response selection process.

Given that all scenarios involved negative effects, the Bane-Boon Theory (Sussman & Oppenheimer, 2020) predicted that the Chain cause with the narrower direct scope would garner lower ratings than the Common Cause condition, which contradicts our finding. Our results also contradict Stephan et al. (2023) experiments, which predicted a "dilution" of causal strength in Common Cause conditions and similar ratings for the target cause in Chain and Control cases due to their identical direct scopes. While our scope manipulation differed slightly from both prior studies (comparing degrees of two rather than three with a single-effect baseline), we find this an unlikely explanation for the discordant results given the generality of the previous researchers' explanations.

One alternative explanation for the enhanced causal strength of 'in-between' causes in canonical Chains is their representation as mechanism nodes (Menzies, 2012), theorized to have an outsized effect on causal intuitions compared with co-variational causes (Ahn et al., 1995; Ahn & Bailenson, 1996; Johnson & Ahn, 2017; Zemla et al., 2017; Russo & Williamson, 2007). While the A→C link in the chain is also part of the mechanistic explanation, the C→B link is the only consistent element across the network types in our study and, therefore, served as the focus of our evaluation. If Menzie's characterization (Menzies, 2012) is accurate, a similar boost in perceived strength should be seen in judgments of the initial cause in a canonical Chain.

A partially divergent explanation is that the middle node in a chain is considered supported by its own cause. This perceived support could arise from violating the normative conditional independence between causes A and B given C. In Chains, the sequence A→B may be perceived as "passing on" some of its causal strength to or suggest greater regularity for the C→B link, compared to A←C link, which would not support the independently produced C→B in a Common Cause network. This line of reasoning leads us to propose what we might term the "Causal Support Hypothesis," which predicts no boost in the perceived strength of the initial cause in a chain (A→C) but enhanced strength for the downstream link from C to B (C→B).

Future experiments can help differentiate between these accounts by incorporating judgments of the initial cause. Direct evaluation of whether the intermediate node is seen as a reliable mechanism would also contribute to this comparison. Better comparisons between generative and preventive

causal chains can further corroborate such findings. Whatever the explanation may be for the biased judgment of intermediate causes in Chains, it seems to be reflected in our language use, as Large Language Models trained predominantly on human-generated text exhibit similar biases.

The implications for domains where causal reasoning is essential, such as medicine and law, could be far-reaching. If LLM-based decision-support systems in these areas inherit biases like the one observed in our study, there is a risk of perpetuating errors, especially at higher model temperatures where biases may be more pronounced and align more closely with human reasoning.

As we increasingly rely on AI for complex decision-making, it becomes more important to study such biases and mitigate them if needed so that more reliable AI systems can aid human decision-makers without introducing undue risk. In the realm of causal cognition, our findings prompt further inquiry into how language shapes or reflects our conceptualization of causality. They suggest that linguistic data, rich with human experiences and inferential patterns, could play a significant role in studying causal biases.

## ACKNOWLEDGEMENTS

AK and UH (Mercator fellowship) were supported by Deutsche Forschungsgemeinschaft (DFG)—Project number 455912038. AK acknowledges the support of Forward College, Berlin. The Bavarian Ministry for Economic Affairs, Regional Development, and Energy supported AK as part of a project to support the thematic development of the Fraunhofer Institute for Cognitive Systems IKS.

BH was supported with funding from the Beckman Institute for Advanced Science and Technology, University of Illinois Urbana-Champaign.

MW and KK acknowledge the support of the German Science Foundation (DFG) project "Causality, Argumentation, and Machine Learning" (CAML2, KE 1686/3-2) of the SPP 1999 "Robust Argumentation Machines" (RATIO). It benefited from the Hessian Ministry of Higher Education, Research, Science and the Arts (HMWK; projects "The Third Wave of AI"). Benefited from the EU Project Tango. Views and opinions expressed are, however, those of the author(s) only and do not necessarily reflect those of the European Union or the European Health and Digital Executive Agency (HaDEA). Neither the European Union nor the granting authority can be held responsible for them. Grant Agreement no. 101120763 - TANGO.

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
