# OpenReview forum: "Biased Causal Strength Judgments in Humans and Large Language Models"
_ICLR.cc/2024/Workshop/Re-Align — ICLR 2024 Workshop Re-Align Poster_

### Official Review · Reviewer_gVym · 2024-02-24
**Review of Submission 74**

**Rating:** 2
**Fit:** 3
**Confidence:** 1

**Workshop Review:**

In this work, the authors investigate the perceived causal strength of humans and large language models (LLMs) by their results comparing three mutually exclusive theories. The main objective of the paper is to establish whether LLMs show similar biases (or similar perceived causal strengths) to humans in causal reasoning.

The paper is clear and full of details on the evaluations and the experiments conducted. All experiments cover two prototypical causal structures with three variables, namely the chain $A \to C \to B$ and the common cause $A \leftarrow C \to B$, and two baselines without the variable $A$. This setup offers the correct ground to test the three different theories, allowing to raise evidence on one over the others, which is a sensible contribution. Finally, the results display similar behavior between human strengths and LLM strengths at different levels of temperature scaling, showing that human biases do also transfer to LLM in standard use situations. This is of interest for the audience at the workshop.

**Reason For Not Giving Higher Score:**

N/A

**Reason For Not Giving Lower Score:**

N/A

**Reviewer Domain:**

machine learning

---

### Official Review · Reviewer_esCv · 2024-02-26
**Biases in probabilistic (causal?) reasoning in humans and LLMs**

**Rating:** 2
**Fit:** 2
**Confidence:** 1

**Workshop Review:**

The paper investigates a bias in the causal judgments of humans and Large Language Models. Firstly, the work compares three different theories or accounts of causal bias ("Dilution", "Boon-Bane" and "Mechanism"). The work then investigates whether LLMs perceive intermediate causes in canonical chain structures as stronger, similar to what was found in humans.

One point I found confusing in this work is that there does not appear to be anything specifically "causal" about the Bayesian networks used in this work, which rather use _probabilistic_ Bayesian networks (Pearl, 1988). Generally, whenever the authors refer to Bayesian theory in the paper, this appears to refer to probabilistic reasoning.

It would be helpful to provide an example (even only in the Appendix) of how subjects should "normatively" answer the questions they are given. This would help better understand the experimental protocol.

The authors describe how they employ "unusual" scenarios to "eliminate [the effect of] prior domain beliefs in participants"; however, it is not fully clear whether participants may be biased by prior beliefs _even regarding such unusual scenarios_. I might be mistaken, but I still have doubts about this aspect of the work.

**Further comments:**

The abstract is, in my view, a bit too long, and it may benefit from shortening.

>  people provide different judgments of a causal Chain, a sequence of causally related events that result in an outcome, than a Common Cause structure where an underlying factor gives rise to multiple effects

> to prevent ceiling effects seen in a pilot study.

These sentences are a bit difficult to parse and unclear.

Section 1.2: The authors never introduce "independence assumptions" up to that point, it would be useful if the term were defined for non-specialist readers.

Moreover, some of the terminology used (e.g., "direct scope of a cause") is slightly non-standard for researchers in causal inference.

**Typos:**

Section 2.1.4: Participants We collected ...

References:

Pearl, Judea. Probabilistic reasoning in intelligent systems: networks of plausible inference. Morgan Kaufmann, 1988.

**Reason For Not Giving Higher Score:**

I want to stress that my evaluation is an educated guess.

One aspect I find problematic is that the work mostly seems to address probabilistic reasoning, as opposed to causal, and it would be worth having more clarity on this in the manuscript. Based on my reading of the workshop's related position paper, the present manuscript may be a good fit for the workshop, though maybe not perfect.

**Reason For Not Giving Lower Score:**

The investigation of biases in causal or probabilistic judgements in humans and machine learning algorithms is interesting and worthwhile.

**Reviewer Domain:**

machine learning

---

### Official Review · Reviewer_VuSb · 2024-03-01

**Rating:** 3
**Fit:** 3
**Confidence:** 1

**Workshop Review:**

This paper explores bias in the causal judgements of humans and LLMs by examining two structures in CBNs: Canonical Chain (A→B→C) and Common Cause (A←B→C) networks. Given B, C is independent of A. Human often ignore such independence. The authors tested this bias with both human studies and LLM experiments.

**Reason For Not Giving Higher Score:**

NA

**Reason For Not Giving Lower Score:**

The paper is well-presented with sufficient experiments to justify the claim and hypotheses. The discussion is also very insightful.

**Reviewer Domain:**

machine learning

---

### Decision · Program_Chairs · 2024-03-02

Accept (Poster)